# Mutual interaction between motor cortex activation and pain in fibromyalgia: EEG-fNIRS study

Eleonora Gentile[1]*, Antonio Brunetti[2], Katia Ricci[1], Marianna Delussi[1], Vitoantonio Bevilacqua[2], Marina de Tommaso[1]

1 Applied Neurophysiology and Pain Unit, SMBNOS Department, Bari Aldo Moro University, Polyclinic General Hospital, Bari, Italy, 2 Department of Electrical and Information Engineering, Polytecnic University of Bari, Bari, Italy

* eleonora.gentile@uniba.it

## Abstract

### Background

Experimental and clinical studies suggested an analgesic effect on chronic pain by motor cortex activation. The present study explored the complex mechanisms of interaction between motor and pain during performing the slow and fast finger tapping task alone and in concomitant with nociceptive laser stimulation.

### Method

The participants were 38 patients with fibromyalgia (FM) and 21 healthy subjects. We used a simultaneous multimodal method of laser-evoked potentials and functional near-infrared spectroscopy to investigate metabolic and electrical changes during the finger tapping task and concomitant noxious laser stimulation. Functional near-infrared spectroscopy is a portable and optical method to detect cortical metabolic changes. Laser-evoked potentials are a suitable tool to study the nociceptive pathways function.

### Results

We found a reduced tone of cortical motor areas in patients with FM compared to controls, especially during the fast finger tapping task. FM patients presented a slow motor performance in all the experimental conditions, requesting rapid movements. The amplitude of laser evoked potentials was different between patients and controls, in each experimental condition, as patients showed smaller evoked responses compared to controls. Concurrent phasic pain stimulation had a low effect on motor cortex metabolism in both groups nor motor activity changed laser evoked responses in a relevant way. There were no correlations between Functional Near-Infrared Spectroscopy (FNIRS) and clinical features in FM patients.

**Data Availability Statement:** Data are available from the SMBNOS, University of Bari on the Open Science Framework at the following link https://osf.

io/d6gq5/?view_only=97e11557adcf49e2b8a
b27a42912bdc7.

**Funding:** The study was supported by Bari Aldo
Moro University Research funds. The funders had
no role in study design, data collection and
analysis, decision to publish, or preparation of the
manuscript.

**Competing interests:** The authors have declared
that no competing interests exist.

## Conclusion

Our findings indicated that a low tone of motor cortex activation could be an intrinsic feature
in FM and generate a scarce modulation on pain condition. A simple and repetitive move-
ment such as that of the finger tapping task seems inefficacious in modulating cortical
responses to pain both in patients and controls. The complex mechanisms of interaction
between networks involved in pain control and motor function require further studies for the
important role they play in structuring rehabilitation strategies.

## Introduction

Fibromyalgia (FM) is a condition of chronic pain [1] whose etiopathogenetic mechanisms are
not yet known. The most typical symptom of FM disease is widespread skeletal muscle pain,
with associated fatigue, alteration of mood, sleep disturbance, cognitive dysfunction [2] and
poor quality of life [3]. Experimental studies have found an analgesic effect on pain induced by
non-invasive brain stimulation techniques such as repetitive transcranial magnetic stimulation
(rTMS) [4] and transcranial direct current stimulation (tDCS) [5–8] on the motor areas. Acti-
vation of the primary motor cortex seems to interact with the cortical regions responsible for
pain processing and have a modulation function on the tM1-thalamic inhibitory networks [9].
Recent evidence indicates an altered functional organization of the primary motor cortex in
subjects suffering from chronic pain [10]. Researchers suggest that motor activity leads to an
improvement in the quality of life of patients [11, 12] so exercise is recommended for the treat-
ment of FM symptoms. Moreover, FM patients have a peculiar limitation of movement that
can manifest itself with dysfunctions in muscle coordination, difficulty in postural control and
reduced speed of motor performance [13, 14]. However, patients suffering from chronic pain
are unlikely to exercise because they fear the worsening of their painful condition [15, 16]. The
exploration of the functional basis of motor cortical areas may be an interesting field to investi-
gate in FM disease.

 Our study aimed to explore the complex mechanisms of interaction between motor and
pain, which have not been yet clearly understood. The co-recording of EEG and functional
Near-Infrared Spectroscopy (fNIRS) has been demonstrated to be a very promising technique
to explore both electrical and metabolic activities [17] during multimodal stimulations condi-
tion. In our preliminary study [18] we adopted a concomitant recording fNIRS and laser
evoked potentials (LEPS) to explore the complex mutual interference between motor cortex
activation and the processing of painful stimuli in FM patients and healthy subjects. The
choice of the multimodal method of EEG-fNIRS simultaneous recording was aimed at explor-
ing the electrophysiological and functional mechanisms underlying the voluntary activation of
cortical areas involved in movement and pain processing one. The advantage of co-registration
lies in being able to obtain functional and electrical data at low cost and with good tolerance to
motor artifacts [19]. Moreover, the light emission in the near-infrared does not contaminate
the electro-physiological signal [20] and vice versa. The principal aim of this paper was to
investigate the motor cortical metabolism and changes of LEPs parameters in FM patients and
healthy subjects. We tested whether there were possible changes induced in motor cortex acti-
vation by laser stimulation and modifications in LEPs during movement tasks.

 The FM patients showed reduced modulation of cortical motor activity during movement
as a probable effect of chronic inhibition. The LEPs amplitude decreased during the movement

task both in patients and controls, though the FM group showed greater internal variability. In the present study, we aimed to enlarge the experimental sample and data analysis to confirm preliminary results [21].

Specific aims were:

1. To compare the changes of haemoglobin activity from the motor cortical regions during the slow and the fast finger tapping task between patients and controls;

2. To compare LEPs changes during slow and fast motor activity between patients and controls;

3. To verify the effects of laser stimulation of the moving hand and the contralateral nonmoving hand on haemoglobin activity

4. To correlate FNIRs/LEPs changes with clinical data in FM group.

## Materials and method

### Subjects

Thirty-eight patients with FM diagnosis and twenty-one healthy subjects served as participants. Diagnosis of FM was in accord to the 2010 American College of Rheumatology criteria, including widespread muscle pain, associated with fatigue, sleep disorders, cognitive impairment, and a number of other physic and psychopathological symptoms [1]. All subjects were right-handed, as confirmed by Edinburgh Handedness Inventory [22]. The experimental procedures of the study were approved by the Ethics Committee of the Bari Polyclinic General Hospital. All the participants signed a written informed consent before inclusion in the study. The exclusion criteria for the recruitment of the study were: less than 8 years of education, any peripheral or central nervous system (CNS) diseases, including spinal cord diseases and radiculopathies, psychiatric diseases, diabetes, active and/or positive history for thyroid insufficiency, renal failure, auto-immune diseases, inflammatory arthritis, systemic connective tissue disease, present or previous history of cancer, as well as use of drugs acting on the CNS or chronic opioid therapy. The FM patients were admitted to the study after their first visit at the Applied Neurophysiology and Pain Unit of Bari University, and before taking the suggested treatment.

The neurologist examined all the patients doing a thorough interview and bedside sensory testing. The FM patients filled out the fibromyalgia impact questionnaire in the Italian version [23] to evaluate their functional status, as recent studies recommend [24]. In all the cases, Self-rating Anxiety Scale [25], Self-Rating Depression Scale [26] and Multidimensional Assessment Fatigue Scale [27] were applied.

Demographic data and clinical features of participants are indicated in Table 1.

**Table 1. Demographic and clinical data of patients and controls groups.**

| Variable | FM patients (N = 38) | Healthy controls (N = 21) |
|---|---|---|
| Age (years) | M = 42,18<br>SD = 10,163 | M = 32,62<br>SD = 13,912 |
| G (M/F) | 3/35 | 8/15 |
| Disease duration (years) | M = 5,48<br>SD = 8,33 | - |
| WPI (0–19) | M = 12,40<br>SD = 4,85 | - |

A, age in years; G, gender; WPI, Widespread Pain Index; M, mean; SD, standard deviation.

## Experimental study design

Participants lied on a comfortable chair in a relaxed state. Before the beginning of the experiment the researcher explained the experimental protocol to each subject. Subjects were invited to perform a finger tapping task, pressing a push-button panel with the right-hand thumb in two modalities, a slow and a fast one. The experimental procedure was based on nine sessions for each subject (Fig 1). Preliminarily we recorded 2 minutes of resting state, during which the participant was requested to stay relaxed with open eyes, fixing on a point on the computer monitor. The subsequent experimental conditions were randomized, and each pre-task baseline was 1 minute in duration. In the laser stimulation condition the participant received laser stimuli on the right- or left-hand dorsum. To keep the participant's attention active the experimenter asked him to count the perceived laser stimuli. Participants were asked to concentrate on the motor task while keeping the rest of their body motionless. The slow finger tapping (SFT) task consisted of pressing a button with the right thumb every 5 seconds following the indications of the experimenter.

The fast finger tapping (FFT) task consisted of clicking a button as quickly as possible. We used the controlled-slow- speed and the maximal-fast- speed the subjects could reach to evaluate the net effect of the movement or of the maximal motor performance on the cortical metabolism.

Both SFT and FFT procedures were repeated during laser stimulation of the right hand (moving hand) and the left–non-moving—hand (subjects performed motor task with the right hand while stimulated on the left one). The laser stimulation of the inactive-left hand served to evaluate the net effect of movement as distractor from painful stimulation.

The speed of the finger tapping tasks was calculated as the number of times per second in which the subject clicked the button on the panel. The interstimulus interval between all the experimental conditions was fixed at 60 s.

## EEG/NIRS recording

The experiment was performed with a co-recording fNIRS-EEG by a compatible cap and a black over-cap to mitigate a possible interference generated by ambient light on the fNIRS signal. We used a continuous wave NIRS system (NIRSport 8X8, Nirx Medical Technologies LLC, Berlin, Germany). The fNIRS data acquisition software was the NIRStar 14.2 (Version 14, Revision 2, Release Build, 2016-04-15 NIRx Medizintechnik GmbH, Berlin, Germany; www.nirx.net).The fNIRS instrument included LED sources and photosensitive detectors (sensitivity: better 1 pW, dynamic range: > 50 dB). Each source employs two LEDs that emit a near-infrared light at 760 nm and 850 nm. The resulting sampling rate of fNIRS signal was 7.81 Hz. The arrangement of sources and detectors resulted in a total of 20 fNIRS measurement channels, 10 for each side of hemisphere (Fig 2). Probes were placed on the motor areas. The inter-optode distance was fixed at 30 mm as, according to previous experimental study [28], this distance was optimal to measure the haemodynamic activity variations over the

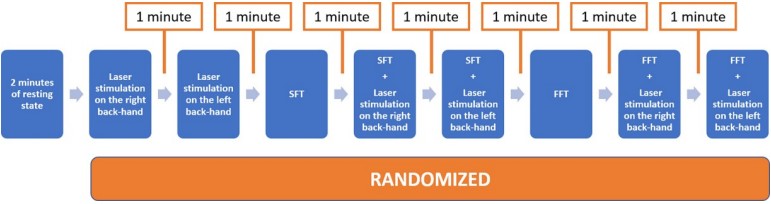

**Fig 1. Design: Randomized sequence of experimental conditions.**

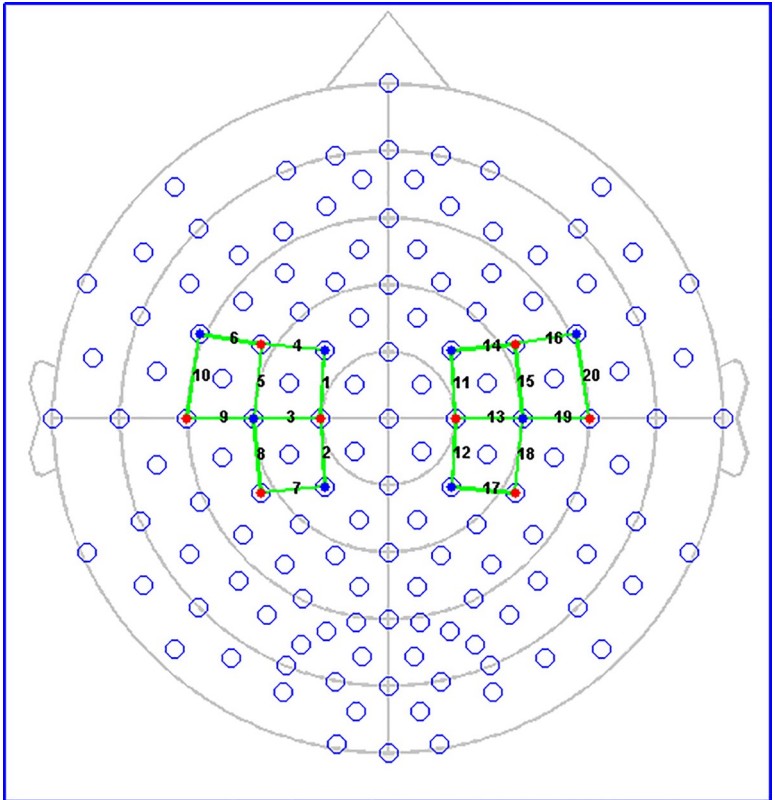

**Fig 2. Channels and optodes configuration.** The red circles indicate sources. The blue circles represent detectors. The green lines show recording channels with the number correspondent.

cerebral surface. Each recording was preceded by a calibration procedure to verify that a good fNIRS signal acquisition was guaranteed. During the calibration procedure the NIRSport instrument determines the signal amplification for each source-detector combination.

EEG data were recorded and amplified using Micromed System Plus (Mogliano Veneto, Italy) at a sampling frequency of 256 Hz. We used a montage with 61 scalp electrodes positioned according to 10–20 International System with reference to the nasion and the ground electrode at the Fpz. Two additional electrodes located above the eyebrows served for electrooculogram recording. The impedance was kept below 5,000 Ω. During the EEG recording, we used digital filters in the 0.1–70 Hz range and a 50 Hz notch filter to allow signal inspection.

## Laser stimulation

Nociceptive stimuli consisted of laser pulses delivered by a $CO_2$ laser (wavelength, 10.6 mm; beam diameter, 2 mm, Neurolas Electronic Engineering Florence, Italy). The interval between each laser stimulus was fixed at 10 s. Patients and controls were stimulated on the back of the hand by laser stimuli of 30 msec duration. We adjusted the laser intensity with the method of the limits, stimulating with laser pulses at an intensity 2.5 Watt above the subjective pain threshold, evaluated on a numeral rating scale (NRS) from 0 to 10, where 4 was the pinprick sensation [29, 30]. The researchers administered a visual analogue scale (VAS) after each laser stimulation to rate the pain intensity perceived by subjects. VAS had values ranging from "0", no pain, with white colour, to "100", intense red, for the worst imaginable pain.

## fNIRS analysis

The fNIRS signal processing method was done with MATLAB (Version R 2018b, MathWorks, Natick, MA, USA) using custom-made scripts with NIRSlab, a commercial software Matlab-based (nirsLAB, version 2017.06, NIRx Medical Technologies, Glen Head, NY, USA). The baseline was defined as the first 120 seconds of the recording. The signal processing was performed by firstly removing discontinuities [31]. Then, according to Remove Spike Artifacts GUI of Nirslab, motion artifacts were removed from the signal. The fNIRS signal was inspected independently by two researchers and the motion artifacts were marked only when they agreed about it. The raw data were digitally filtered in the band-pass 0.005–0.2 Hz to remove low oscillations, like respiratory and cardiac frequencies from fNIRS signal. The spectrum as published by W.B. Gratzer (Med. Res. Council Labs, Holly Hill, London and N. Kollias, Wellman Laboratories, Harvard Medical School, Boston, MA, USA) was selected for the molar extinction coefficients of haemoglobin. Optical intensity measurements were converted to oxyhaemoglobin ($\Delta HbO_2$) and deoxyhaemoglobin ($\Delta Hb$) concentration by the modified Beer-Lambert law [32, 33]. The unit of haemoglobin concentration is measured in mmol per liter (mmol/liter). The mean values of the haemoglobin concentration were subtracted to calculate the changes in $\Delta HbO_2$ and $\Delta Hb$ during the experimental tasks. To subtract the baseline, the range of timeframes that indicated the rest status was entered for each individual subject in the NIRSlab software before the application of the modified Beer-Lambert law.

We performed a baseline correction before calculating the mean $\Delta HbO_2$ and $\Delta Hb$ concentrations in the different experimental conditions. The $\Delta HbO_2$ and $\Delta Hb$ levels in the moving situation, subtracting the resting state in the 2 min preceding fast and slow motor task (Fig 1), were the variables considered for the comparison between groups.

## LEPs analysis

To analyse the EEG signal we used an open-source Matlab toolbox named Letswave 6 (André Mouraux, Brussels, Belgium; www.letswave.org). The pre-processing signal method consisted of frequency filtering, bad electrodes interpolation, segmentation in epochs, artefact rejection, independent component analysis (ICA) decomposing method for ocular artifacts. The Butterworth IIIR filter was applied for bandpass filtering in the 0.01–30 Hz range. Bad channels were removed with subsequent interpolation. The motor artifacts were visually inspected and removed. We applied ICA method to remove ocular and motor artifacts from the EEG signal. We averaged the EEG epochs in the 100 msec preceding and 1000 msec following laser stimuli. We examined in the current study the N1, an early component detected on the contralateral temporal regions at the stimulation side (T3 or T4 channel), and the N2 and P2 vertex waves (late component) recorded on the Cz electrode [34, 29]. The waves amplitude was computed from the baseline. Latencies were measured from the 0 time to the maximal amplitude of each wave.

## Statistical analysis

Statistical data analysis was performed using IBM SPSS Statistics software, version 21. For all statistical tests a $p$-value lower than 0.05 was considered statistically significant.

A two-way analysis of variance (ANOVA) corrected for age was used for the comparison of finger tapping speed between groups.

## FNIRS

For topographical analysis, we used the Statistical Parameter Mapping NIRS-SPM (SPM 8) tool implemented in NIRSlab (version 2017.6), modelled with the Generalized Linear Model

(GLM), to identify the brain regions active during the execution of the tasks in the single cases. We considered the Haemodynamic Response Function (HRF) to model the haemodynamic response under the experimental tasks in the Statistical Parametric Mapping (SPM1- within subject) analysis [35], computing the degree of activation on each channel in respect to the baseline (beta value). Repeated-measures ANOVA tests for each channel were performed, considering the beta values during each experimental condition as within-subject-factors and groups as a between-subjects factor. Then the SPM 2 (between subjects) analysis was performed to identify the fNIRS channels where both HbO and HbR changed in a significant way in the finger tapping tasks between groups (p< 0.05 corrected for multiple comparisons).

The Lateralization Index (LI) was used to estimate the hemispheric dominance between the right and left motor areas. According to Arun et al. [36], we used the following equation:

$$LI = \frac{\max(\Delta HbO_{2\_L}) - \max(\Delta HbO_{2\_R})}{\max(\Delta HbO_{2\_L}) + \max(\Delta HbO_{2\_R})}$$

Where $\Delta HbO_{2\_L}$ and $\Delta HbO_{2\_R}$ are the maximum values for $\Delta HbO_2$ concentration changes in the channels on the left and right hemisphere respectively. In this case, we evaluated the LI for the channels 4, 6, 10 which showed significant activation during the task in the comparison between groups.

We used the beta values obtained in the SPM 1 analysis for calculating the LI. Since these parameters may have negative values, we used a modified formula for computing the LI, as reported in [37]. The modified equation is herein reported:

$$LI = \frac{\Delta HbO_{2\_L} - \Delta HbO_{2\_R}}{|\Delta HbO_{2\_L}| + |\Delta HbO_{2\_R}|}$$

Laser evoked potentials. We preliminary ran out a repeated-measures ANOVA test with the LEP features as within-subject-factors and groups as a between-subjects factor, using the above-mentioned SPSS software. Individual univariate repeated contrasts were also applied. For topographical representation of LEP statistical analysis, we used the above described LETSWAVE MATLAB software, implementing a parametric statistic with groups and conditions as factors, and Bonferroni correction for multiple comparisons.

## Correlations

The Person's correlation coefficient was performed to evaluate a possible linear correlation between the LEPs and the fNIRS signals and clinical features, using a script executed in IBM SPSS.

## Results

Finger tapping was slower in FM patients as compared to controls, independently from the concurrent laser stimulation (Table 2, Fig 3).

## fNIRS results

The main difference in activation between the two groups, regardless of the experimental condition, was evident in correspondence of the left motor cortex.

Below we show the detailed results of the repeated measures for every single statistically significant channel. Table 3 and Fig 4A show the results of the pairwise comparisons for $\Delta HbO_2$ between the groups for each condition considering the channel number 4.

**Table 2. Results of one-way ANOVA.**

| Test between subject effects | | | | | |
|---|---|---|---|---|---|
| Dependent variable: speed | | | | | |
| Source | Sum of squares | Df | Mean of squares | F | Sig. |
| Correct model | 719,871[b] | 5 | 143,974 | 11,066 | ,000 |
| Intercept | 93951,014 | 1 | 93951,014 | 7221,312 | ,000 |
| Group | 693,381 | 1 | 693,381 | 53,295 | ,000 |
| Condition | 16,304 | 2 | 8,152 | ,627 | ,536 |
| Group * Condition | 5,375 | 2 | 2,688 | ,207 | ,814 |
| Error | 2146,690 | 165 | 13,010 | | |
| Total | 105771,722 | 171 | | | |
| Corrected total | 2866,561 | 170 | | | |

Speed: variable; Groups and conditions (FFT, FFT during laser stimulation on the right hand, FFT during laser stimulation on the left hand): factors.

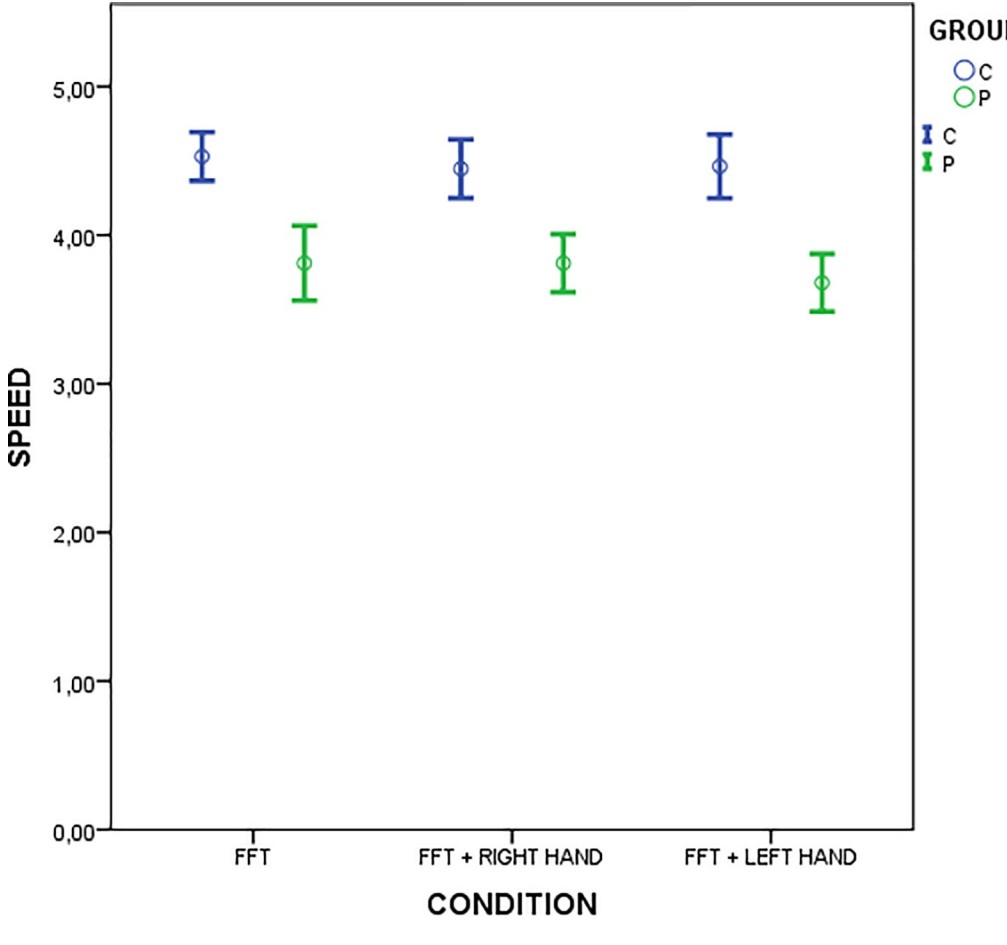

**Fig 3.** Mean values of finger tapping speed in motor task conditions in patients (green) and controls (blue). Statistical comparison is reported in Table 2.

**Table 3. Results of pairwise comparisons between groups C (controls) and P (FM patients) for each experimental condition in channel 4 for $\Delta HbO_2$.**

Pairwise Comparisons

Measure: Ch4_ $\Delta HbO_2$

| Condition | (I) GROUP | (J) GROUP | Mean Difference (I-J) | Std. Error | Sig.[b] | 95% Confidence Interval for Difference[b] | |
|---|---|---|---|---|---|---|---|
| | | | | | | Lower Bound | Upper Bound |
| Resting state | C | P | -1,953E-007 | ,000 | ,992 | -3,827E-005 | 3,788E-005 |
| SFT | C | P | 7,586E-005 | ,000 | ,134 | -2,417E-005 | ,000 |
| FFT | C | P | 7,448E-005* | ,000 | ,031* | 7,245E-006 | ,000 |
| SFT + Laser on the left hand | C | P | 2,190E-005 | ,000 | ,408 | -3,074E-005 | 7,455E-005 |
| SFT + Laser on the right hand | C | P | 1,008E-005 | ,000 | ,695 | -4,108E-005 | 6,124E-005 |
| FFT + Laser on the left hand | C | P | 5,140E-005* | ,000 | ,046* | 8,543E-007 | ,000 |
| FFT + Laser on the right hand | C | P | 3,703E-005 | ,000 | ,179 | -1,746E-005 | 9,152E-005 |

Based on estimated marginal means

*. The mean difference is significant at the, 05 level.

[b]. Adjustment for multiple comparisons: Bonferroni.

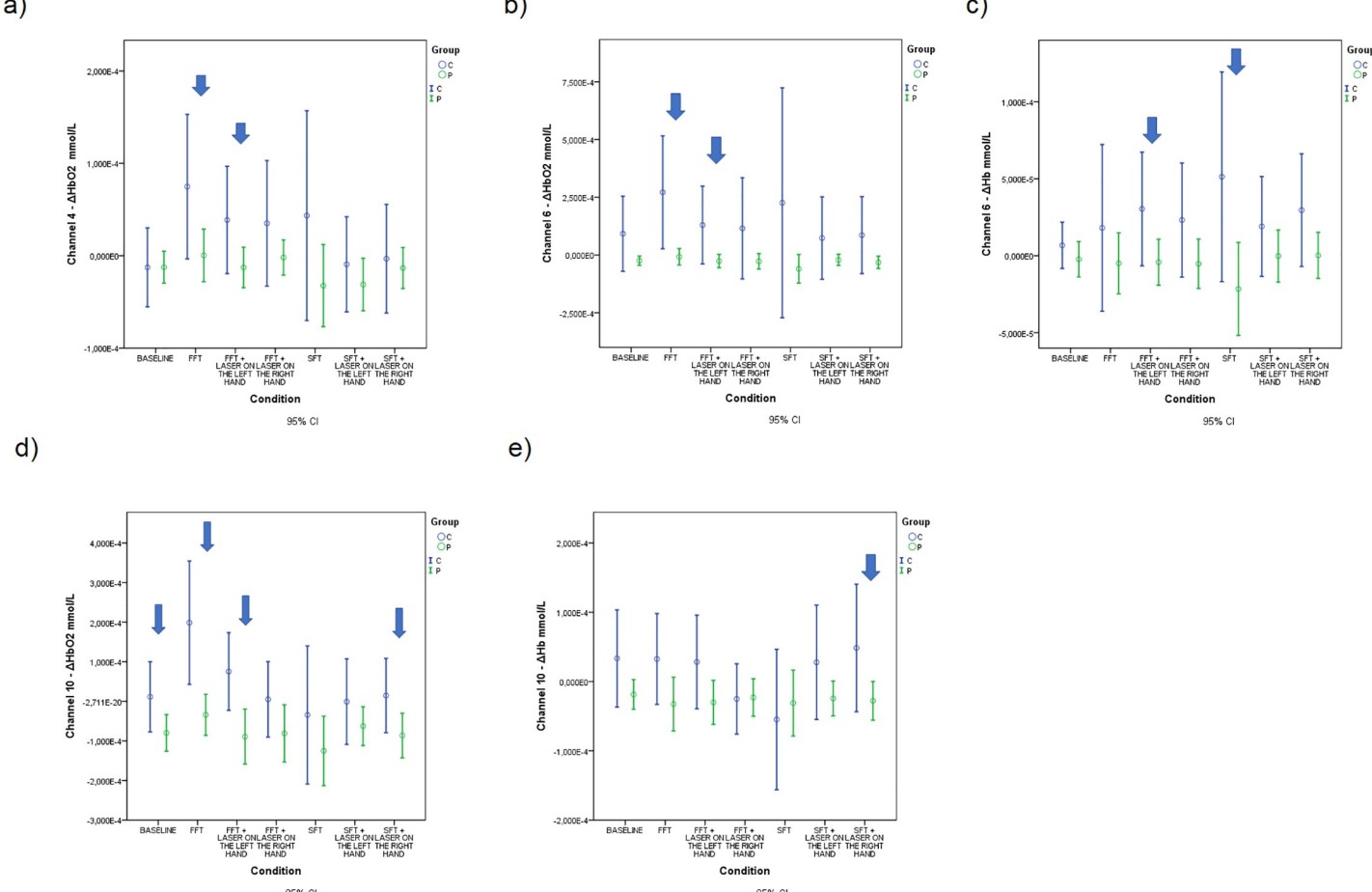

**Fig 4. Estimated means of $\Delta HbO_2$ or $\Delta Hb$ expressed in mmol/L for significant channels in different conditions.** (a) Estimated means of $\Delta HbO_2$ for channel 4; (b) Estimated means of $\Delta HbO_2$ for channel 6; (c) Estimated means of $\Delta Hb$ for channel 6; (d) Estimated means of $\Delta HbO_2$ for channel 10; (e) Estimated means of $\Delta Hb$ for channel 10.

**Table 4. Results of pairwise comparisons between groups for each experimental condition in channel 6 for ΔHbO2.**

**Pairwise Comparisons**

Measure: Ch6_ $\Delta HbO_2$

| Condition | (I) GROUP | (J) GROUP | Mean Difference (I-J) | Std. Error | Sig.[b] | 95% Confidence Interval for Difference[b] | |
|---|---|---|---|---|---|---|---|
| | | | | | | Lower Bound | Upper Bound |
| Resting state | C | P | ,000 | ,000 | ,053 | -1,775E-006 | ,000 |
| SFT | C | P | ,000 | ,000 | ,120 | -7,640E-005 | ,001 |
| FFT | C | P | ,000* | ,000 | ,003* | 9,924E-005 | ,000 |
| SFT + Laser on the left hand | C | P | 9,473E-005 | ,000 | ,152 | -3,584E-005 | ,000 |
| SFT + Laser on the right hand | C | P | ,000 | ,000 | ,061 | -5,770E-006 | ,000 |
| FFT + Laser on the left hand | C | P | ,000* | ,000 | ,016* | 3,072E-005 | ,000 |
| FFT + Laser on the right hand | C | P | ,000 | ,000 | ,082 | -1,895E-005 | ,000 |

Based on estimated marginal means

*. The mean difference is significant at the, 05 level.

[b]. Adjustment for multiple comparisons: Bonferroni.

Table 4 and Fig 4B show the results of the pairwise comparisons for $\Delta HbO_2$ between the groups for each condition considering the channel number 6.

Table 5 and Fig 4C show the results of the pairwise comparisons for $\Delta Hb$ between the groups for each condition considering the channel number 6.

Table 6 and Fig 4D show the results of the pairwise comparisons for $\Delta HbO_2$ between the groups for each condition considering the channel number 10.

Table 7 and Fig 4E show the results of the pairwise comparisons for $\Delta Hb$ between the groups for each condition considering the channel number 10.

Considering the results obtained for fNIRS, we designed the F-contrast between groups plotting the F-values for all the channels during the baseline condition, FFT and SFT conditions, for both $\Delta HbO_2$ and $\Delta Hb$.

Fig 5A shows the topographical maps with the F-contrast comparison between the FM and control groups for the resting state condition. The significant changes in oxyhaemoglobin levels between groups were on the channel 10 for $\Delta HbO_2$.

**Table 5. Results of pairwise comparisons between groups for each experimental condition in channel 6 for ΔHb.**

**Pairwise Comparisons**

Measure: Ch6_ $\Delta Hb$

| Condition | (I) GROUP | (J) GROUP | Mean Difference (I-J) | Std. Error | Sig.[b] | 95% Confidence Interval for Difference[b] | |
|---|---|---|---|---|---|---|---|
| | | | | | | Lower Bound | Upper Bound |
| Resting state | C | P | 9,019E-006 | ,000 | ,338 | -9,654E-006 | 2,769E-005 |
| SFT | C | P | 7,290E-005* | ,000 | ,024* | 1,001E-005 | ,000 |
| FFT | C | P | 2,304E-005 | ,000 | ,326 | -2,355E-005 | 6,964E-005 |
| SFT + Laser on the left hand | C | P | 1,919E-005 | ,000 | ,238 | -1,303E-005 | 5,141E-005 |
| SFT + Laser on the right hand | C | P | 2,936E-005 | ,000 | ,078 | -3,356E-006 | 6,208E-005 |
| FFT + Laser on the left hand | C | P | 3,461E-005* | ,000 | ,040* | 1,715E-006 | 6,751E-005 |
| FFT + Laser on the right hand | C | P | 2,840E-005 | ,000 | ,099 | -5,487E-006 | 6,229E-005 |

Based on estimated marginal means

*. The mean difference is significant at the, 05 level.

[b]. Adjustment for multiple comparisons: Bonferroni.

**Table 6. Results of pairwise comparisons between groups for each experimental condition in channel 10 for $\Delta HbO_2$.**

Pairwise Comparisons

Measure: Ch10_ $\Delta HbO_2$

| Condition | (I) GROUP | (J) GROUP | Mean Difference (I-J) | Std. Error | Sig.[b] | 95% Confidence Interval for Difference[b] | |
|---|---|---|---|---|---|---|---|
| | | | | | | Lower Bound | Upper Bound |
| Resting state | C | P | 9,102E-005* | ,000 | ,033* | 7,611E-006 | ,000 |
| SFT | C | P | 9,088E-005 | ,000 | ,264 | -7,029E-005 | ,000 |
| FFT | C | P | ,000* | ,000 | ,000* | ,000 | ,000 |
| SFT + Laser on the left hand | C | P | 6,182E-005 | ,000 | ,198 | -3,313E-005 | ,000 |
| SFT + Laser on the right hand | C | P | ,000* | ,000 | ,040* | 4,944E-006 | ,000 |
| FFT + Laser on the left hand | C | P | ,000* | ,000 | ,005* | 5,290E-005 | ,000 |
| FFT + Laser on the right hand | C | P | 8,595E-005 | ,000 | ,136 | -2,792E-005 | ,000 |

Based on estimated marginal means

*. The mean difference is significant at the, 05 level.

[b]. Adjustment for multiple comparisons: Bonferroni.

Concerning the FFT condition, Fig 5B shows the F-contrast comparison between the FM and control groups. The higher difference between the groups is located on channels 4, 6 and 10 for $\Delta HbO_2$ as confirmed by the results of Repeated Measures ANOVA.

Fig 5C shows the F-contrast comparison between the FM and control groups for the FFT during concomitant laser stimulation on the left-hand condition. As confirmed by the Repeated Measures ANOVA analysis, the higher difference between the groups was located on channel 4, 6 and 10 for $\Delta HbO_2$ and on channel 6 for $\Delta Hb$.

Fig 5D shows the F-contrast comparison between the FM and control groups for the SFT condition. As confirmed by the Repeated Measures ANOVA analysis, the higher difference between the groups was located on channel 6 for $\Delta Hb$.

Finally, Fig 5E shows the F-contrast comparison between the FM and control groups for the SFT condition. As confirmed by the Repeated Measures ANOVA analysis, the higher difference between the groups was located on channel 6 for $\Delta Hb$.

**Table 7. Results of pairwise comparisons between groups for each experimental condition in channel 10 for $\Delta Hb$.**

Pairwise Comparisons

Measure: Ch10_$\Delta Hb$

| Condition | (I) GROUP | (J) GROUP | Mean Difference (I-J) | Std. Error | Sig.[b] | 95% Confidence Interval for Difference[b] | |
|---|---|---|---|---|---|---|---|
| | | | | | | Lower Bound | Upper Bound |
| Resting state | C | P | 5,206E-005 | ,000 | ,054 | -8,633E-007 | ,000 |
| SFT | C | P | -2,379E-005 | ,000 | ,600 | ,000 | 6,664E-005 |
| FFT | C | P | 6,508E-005 | ,000 | ,055 | -1,439E-006 | ,000 |
| SFT + Laser on the left hand | C | P | 5,221E-005 | ,000 | ,099 | -1,009E-005 | ,000 |
| SFT + Laser on the right hand | C | P | 7,635E-005* | ,000 | ,031* | 7,033E-006 | ,000 |
| FFT + Laser on the left hand | C | P | 5,839E-005 | ,000 | ,058 | -2,002E-006 | ,000 |
| FFT + Laser on the right hand | C | P | -2,019E-006 | ,000 | ,934 | -5,044E-005 | 4,640E-005 |

Based on estimated marginal means

*. The mean difference is significant at the, 05 level.

[b]. Adjustment for multiple comparisons: Bonferroni.

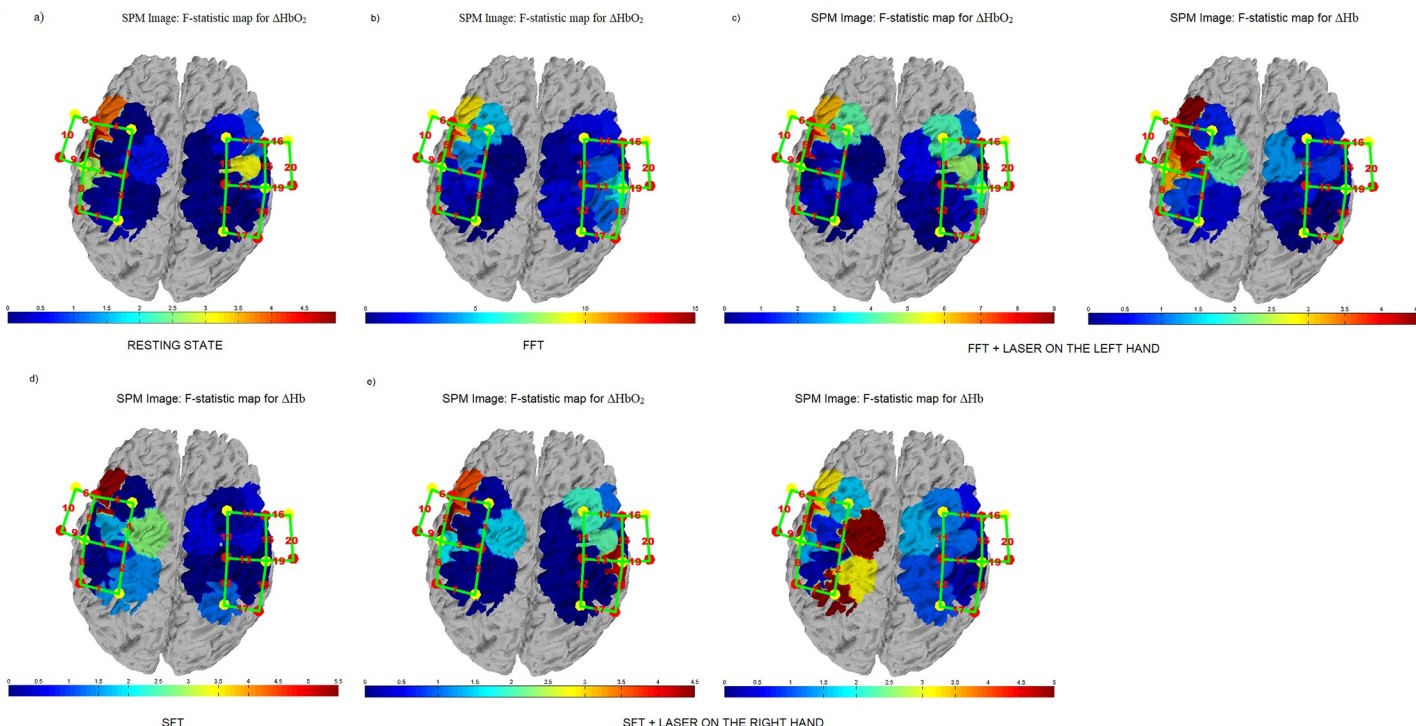

**Fig 5. F-statistic values of ΔHbO$_2$ and ΔHb during different conditions. FM and Control groups activation maps using canonical HRF model. The higher difference between control subjects and patients' activations is represented with the red colour.** (a) F-statistic values of ΔHbO$_2$ during the resting state condition; (b) F-statistic values of ΔHbO$_2$ during the FFT condition; (c) F-statistic values of ΔHbO$_2$ and ΔHb during the FFT + LASER ON THE LEFT-HAND condition; (d) F-statistic values of ΔHb during the SFT condition; (e) F-statistic values of ΔHbO$_2$ and ΔHb during the SFT + LASER ON THE RIGHT-HAND condition.

Regarding the evaluation of the lateralization during the tasks, we obtained the following results, considering a threshold for lateralization (L$_{TH}$) of 0.15 [37]. Specifically, if LI > L$_{TH}$, the subject was considered left dominant; if LI < -L$_{TH}$, the subject was right dominant; if |LI| < L$_{TH}$, the subject had a bilateral dominance. During the FFT task the 76,19% of control subjects and the 76,31% of FM patients were left predominant.

## LEPs results

Data of LEPs (latency and amplitude) are reported in S1 Table for patients and S2 Table for controls.

Fig 6 shows group-level average LEPs in the experimental conditions with laser stimulation on the right hand.

Fig 7 shows group-level average LEPs in the experimental conditions with laser stimulation on the left hand.

For almost all the LEPs parameters both in the patient group and in the control group there were no statistically significant changes between the different experimental conditions. The detailed results will be shown below (Figs 6 and 7).

The N1 and N2P2 amplitude **was significantly** smaller in patients than controls when the stimulation was on the right hand (Table 8, Fig 8).

Moreover, we observed a significant difference in N1 latency between groups for experimental condition of FFT task during laser stimulation on the right hand (Table 9, Fig 9).

We observed no significant changes in LEPs parameters when the stimulation was on the left hand independent from experimental condition.

a)

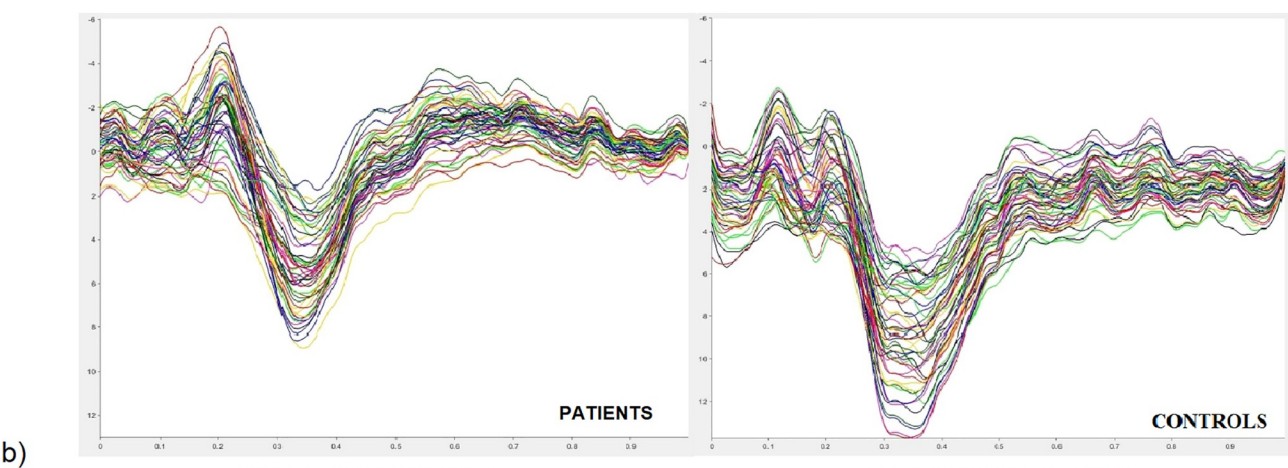

LASER ON THE RIGHT HAND

LASER ON THE RIGHT HAND

b)

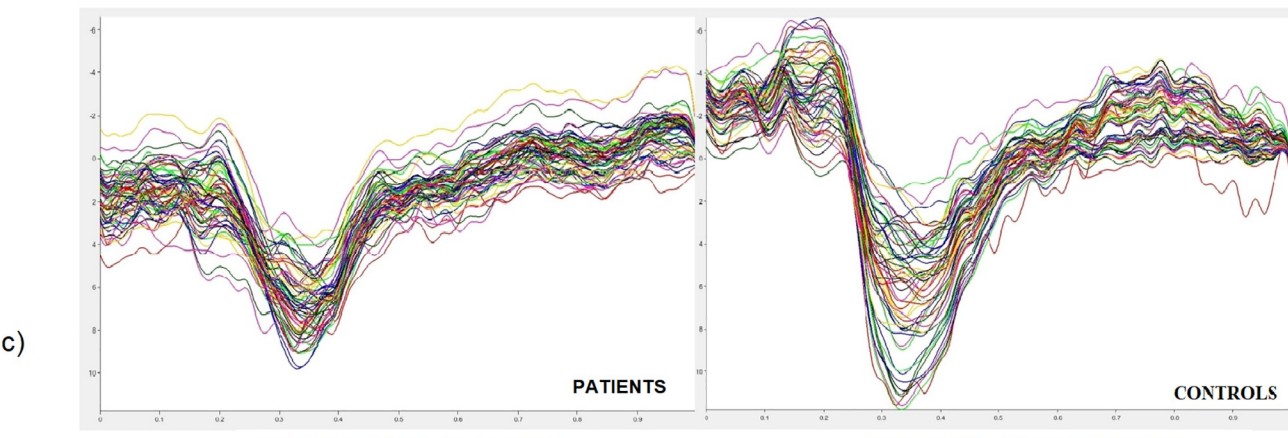

SFT+ LASER ON THE RIGHT HAND

SFT+ LASER ON THE RIGHT HAND

c)

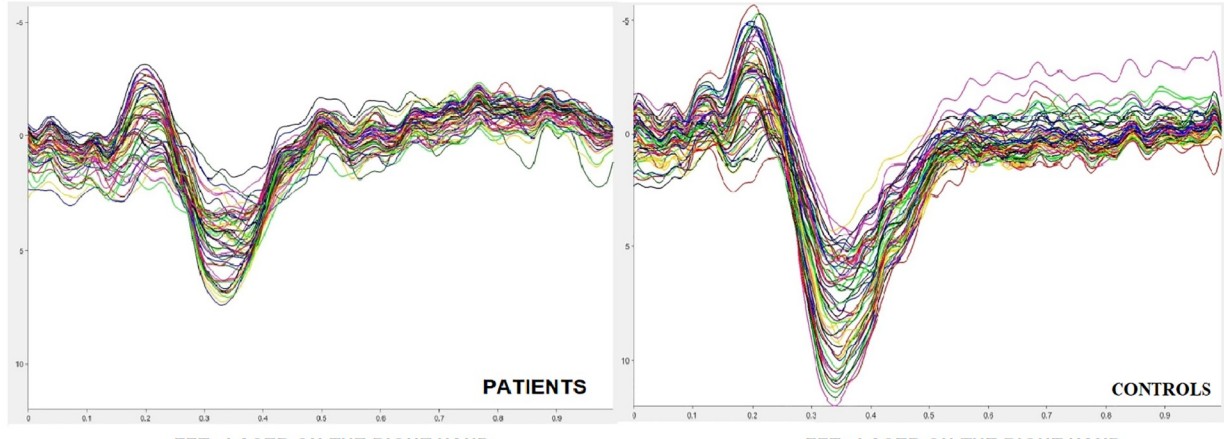

FFT+ LASER ON THE RIGHT HAND

FFT+ LASER ON THE RIGHT HAND

**FFT + LASER ON THE RIGHT HAND**

**FFT + LASER ON THE RIGHT HAND**

**Fig 6. Grand average of LEPs by right hand stimulation in patients and controls.** (a) laser on the right hand, (b) SFT task during concomitant stimulation on the right hand, (c) FFT task during concomitant stimulation on the right hand both in patients and controls.

Intensity of pain perception. VAS values were similar in basal condition and during motor tasks. However, we found a significant difference in the intensity of pain perception between the group of patients and the group of controls. For details see Table 10.

## Correlation results

The linear regression analysis was performed between the amplitude of LEPs and haemodynamic activity for each experimental condition. Our results did not indicate any significant correlation between these data in both patient and control groups. Moreover, we observed no significant correlations, or significant but low statistical level correlations, comparing fNIRS data and LEPs features with clinical characteristics of patients, as disease duration expressed in years, WPI [38], Self-rating Anxiety Scale [25], Self-Rating Depression Scale [26], Multidimensional Assessment Fatigue Scale [27] (S3–S9 Tables).

Moreover, the same results were obtained analysing the correlations between finger tapping speed and haemodynamic responses in each channel in both patients and controls group (S10–S12 Tables).

## Discussion

The main results of the present study partly confirmed previous findings [21]. Patients suffering from FM had a reduced motor performance as tested by finger tapping task, and a reduced tone of cortical motor areas, especially evident during fast movement. Concurrent phasic pain stimulation had limited effect on motor cortex metabolism in both groups, nor the motor activity changed the laser evoked responses in a relevant way. The reduced tone of motor areas activation was independent of FM duration and severity. In the following paragraphs main results are discussed in detail.

### Reduced motor performance and motor cortical areas activation in FM patients

The slow motor performance expressed by FM patients during finger tapping was present in all the experimental conditions that requested a rapid movement and independently of the laser stimulation on the active or on the inactive hand. A meta-analysis, conducted on functional neuroimaging studies [39], indicates the frequent use of the finger tapping task to investigate the functioning of the motor cortex, as it is a simple task to be performed both for patients with motor difficulties and healthy subjects. For this reason, we have chosen this type of motor task as it was appropriate for motor activation of FM patients. In a study on the evaluation of the kinematic parameters of gait and balance, patients with FM showed impaired motor performance [40]. We hypothesize that FM patients had a lower speed of finger tapping task than controls due to the interaction of several factors linked to pain condition. Moreover, studies focused on motor ability suggest that FM patients had low manual dexterity [41] and minor handgrip strength [13] compared to healthy subjects. In a study on the evaluation of the kinematic parameters of gait and balance, patients with FM showed impaired motor performance [40]. The low motor performance of FM patients could be due to fear of movement [42] or cognitive problems with impaired motor programming [43–44]. It is possible that the reduced speed of information processing which often characterizes patients with chronic pain can also affect the control and speed of motor responses [45]. The motor impairment could be

a)

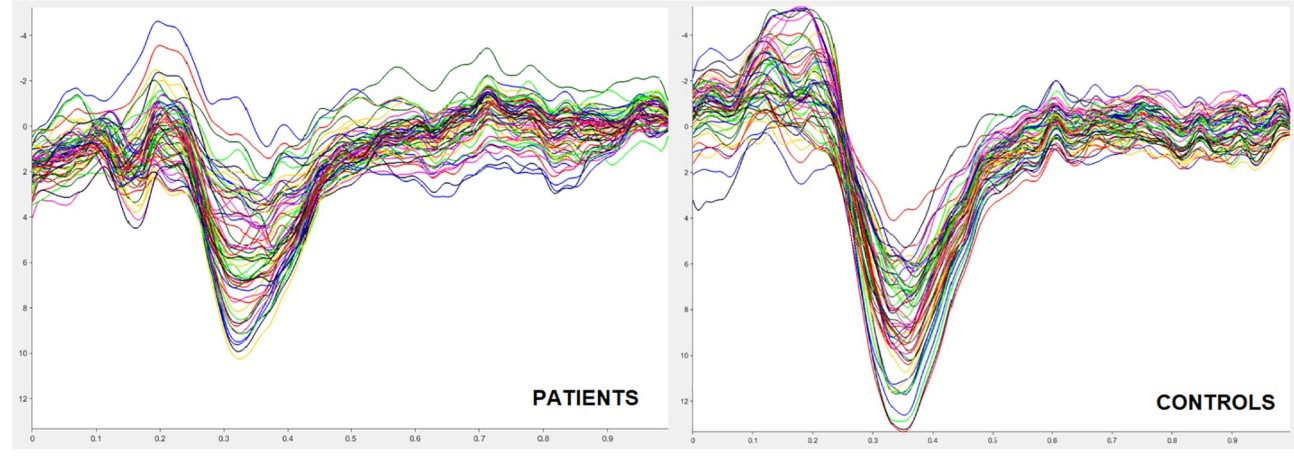

**LASER ON THE LEFT HAND**

**LASER ON THE LEFT HAND**

b)

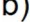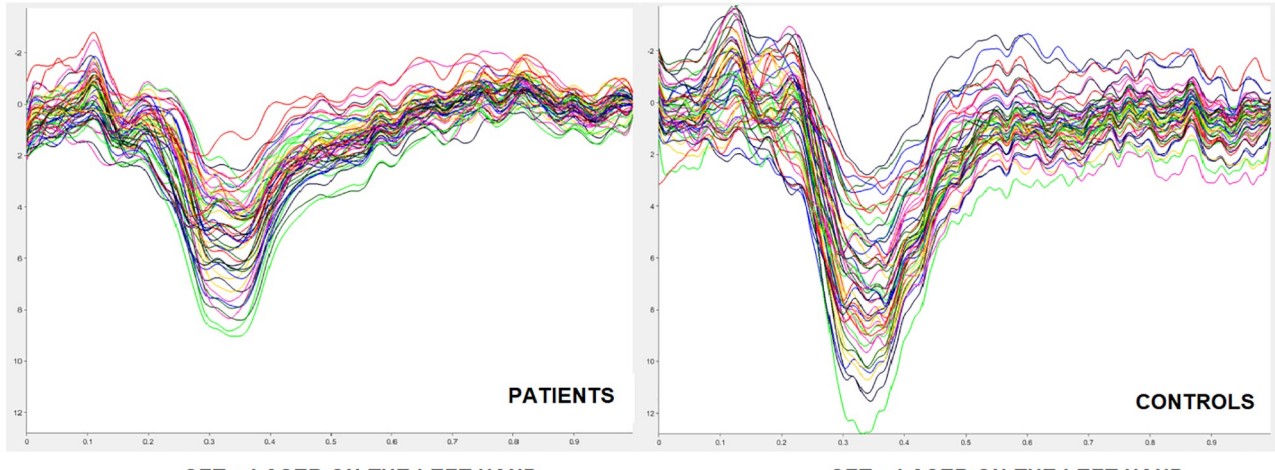

**SFT + LASER ON THE LEFT HAND**

**SFT + LASER ON THE LEFT HAND**

c)

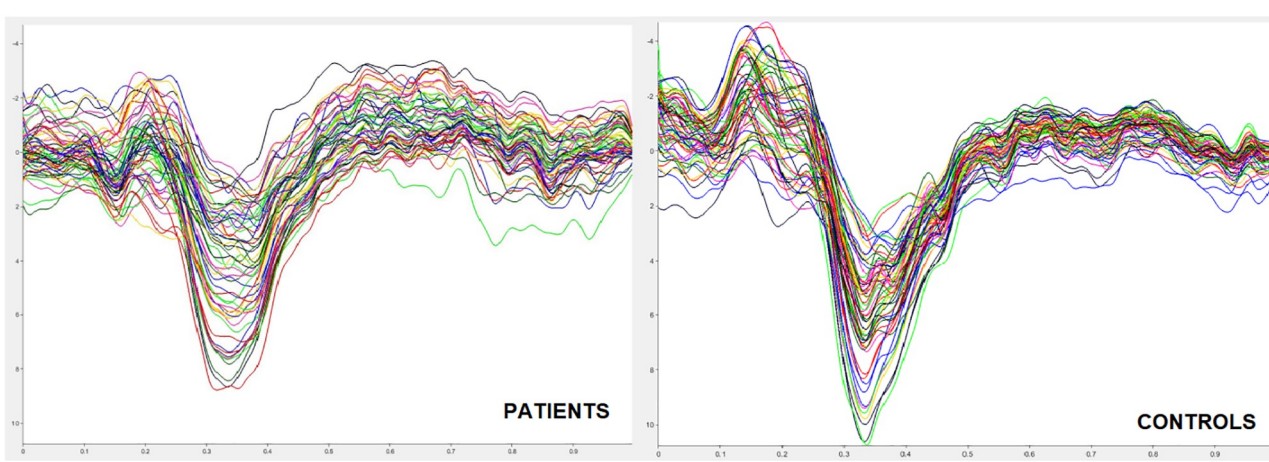

**FFT + LASER ON THE LEFT HAND**

**FFT + LASER ON THE LEFT HAND**

**Fig 7. Grand average of LEPs by left hand stimulation in patients and controls.** (a) laser on the left hand, (b) SFT task during concomitant stimulation on the left hand, (c) FFT task during concomitant stimulation on the left hand) both in patients and controls.

a constitutional tract in FM, as it seemed independent from disease severity and duration (S3–S9 Tables). Further study would evaluate the evolution of this motor dysfunction and the possible effect of current treatments [46].

As we expected, the spatial distribution of brain activity during the movement of the right hand involved the left prefrontal regions, corresponding to the primary and supplementary motor cortex. In the resting state patients and controls showed a significantly different activation in channel 10, with a trend toward a greater level of concentrations of $\Delta HbO_2$ in healthy subjects. Furthermore, our results indicated that there were significant differences in motor cortical activation between patients and controls during the fast movement condition on channels 4, 6, 10. We did not observe a compensatory activity of right hemisphere in FM patients, as generally occurs in unilateral motor cortex dysfunction [47] The hypometabolism here observed in FM patients could involve the bilateral cortical motor areas, with an absence of contralateral compensation during simple unilateral motor activities. We can assume that the activation levels of the motor cortex were independent of the velocity of the finger tapping task, as, in line with our previous study [21], we did not find any relevant correlation between the motor speed and haemodynamic responses either in patients or in controls. The results of haemodynamic responses suggest that FM patients could have a dysfunction in supplementary and primary motor cortex modulation. In this regard we can suppose that a possible altered cortical motor function could characterize this chronic pain syndrome. Patients did not show any modulation of haemoglobin levels during the concurrent laser stimulation, confirming a rigid modality of motor cortical activation. Scientific evidences suggest a complex mechanism of reorganization of the motor cortex in conditions of chronic pain, whose functioning is not yet clear [48]. Repetitive TMS [49] and fMRI [50] studies documented that patients suffering from chronic pain presented with altered spinocortical and intracortical excitability of primary motor cortex, that could contribute to the impairment of their motor performance and the limited modulation of chronic symptoms related to the pain condition [51]. Recent studies [52] on animal models, confirmed that the repetitive stimulation of motor cortex is able to modify synaptic connections involved in pain control, with an adjustment of mechanical hypersensitivity occurring in neuropathic pain. Activation of the motor cortex has an analgesic effect on pain conditions [53], but the motor cortical dysfunction that seemed to characterize FM patients could reduce the modulating effect on pain.

The gap of cortical metabolism characterizing patients in respect to controls during fast movement, was lost in the condition of concurrent fast finger tapping and laser stimulation of the right active hand. This phenomenon could be based on a partial inhibition of cortical

**Table 8. Repeated Measures ANOVA for N1 and N2P2 amplitude.**

**Tests of Between-Subjects Effects**

**Measure: Amplitude N1**

| Source | Type III Sum of Squares | df | Mean Square | F | Sig. |
|---|---|---|---|---|---|
| Group | 542,752 | 1 | 542,752 | 8,290 | ,006* |

**Measure: Amplitude N2P2**

| Source | Type III Sum of Squares | df | Mean Square | F | Sig. |
|---|---|---|---|---|---|
| Group | 1365,861 | 1 | 1365,861 | 6,133 | ,016* |

Significant Group in Repeated Measures ANOVA.

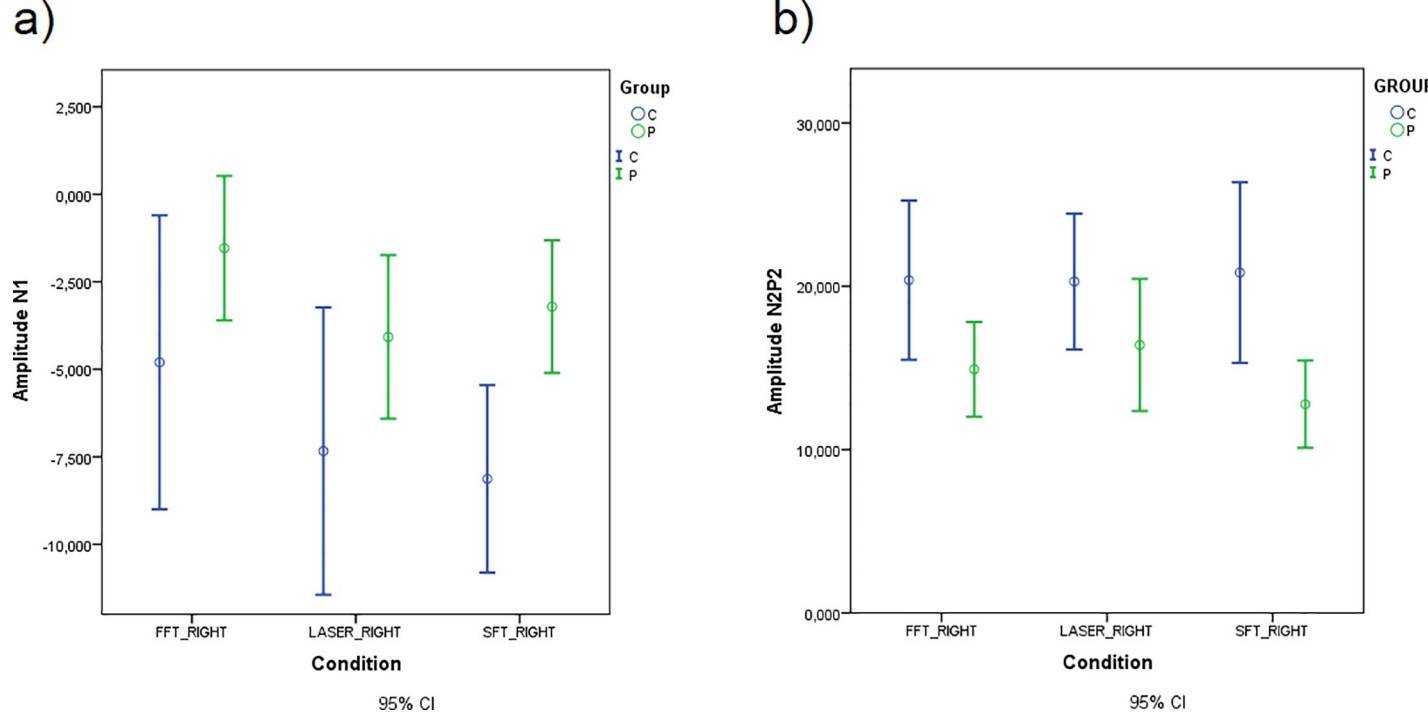

**Fig 8. Mean of N1 and N2P2 amplitude in experimental conditions with laser stimulation on the right hand for patients and controls.** (a) N1 amplitude. (b) N2P2 amplitude.

motor areas during concurrent nociceptive stimulation in healthy controls, generating a loss of the metabolic advantage in comparison to patients. The laser stimulation could exert a modulation effect on the motor areas activation in controls when the task requires more effort as in the case of fast finger tapping. A similar phenomenon emerged during the concurrent laser stimulation of the non-moving hand, though it was not as relevant as for the right hand. The sensory motor integration of proprioceptive inputs coming from the moving hand, could further interfere with the laser stimuli and reinforce the inhibition they could exert on the metabolism of the motor cortical areas. Recent studies on metabolic changes of cortical areas measured by proton magnetic resonance spectroscopy in patients with chronic low back pain, indicated alteration in the biochemical profile of several cortical areas, including the motor cortex [54]. In sum, the general tone of motor cortex activation is lower in FM patients as compared to controls, especially during fast movement. Considering that in healthy subjects the concurrent laser stimulation of the moving hand reduced the haemoglobin changes during the

**Table 9. Repeated measures ANOVA for N1 latency.**

**Univariate Tests**

**Measure: LATENCY_N1**

| Condition | | Sum of Squares | Df | Mean Square | F | Sig. |
|---|---|---|---|---|---|---|
| FFT + LASER ON THE RIGHT HAND | Contrast | ,002 | 1 | ,002 | 6,160 | ,016 |
| | Error | ,013 | 51 | ,000 | | |

Each F tests the simple effects of GROUP within each level combination of the other effects shown. These tests are based on the linearly independent pairwise comparisons among the estimated marginal means.

Significant GROUP x Condition interactions.

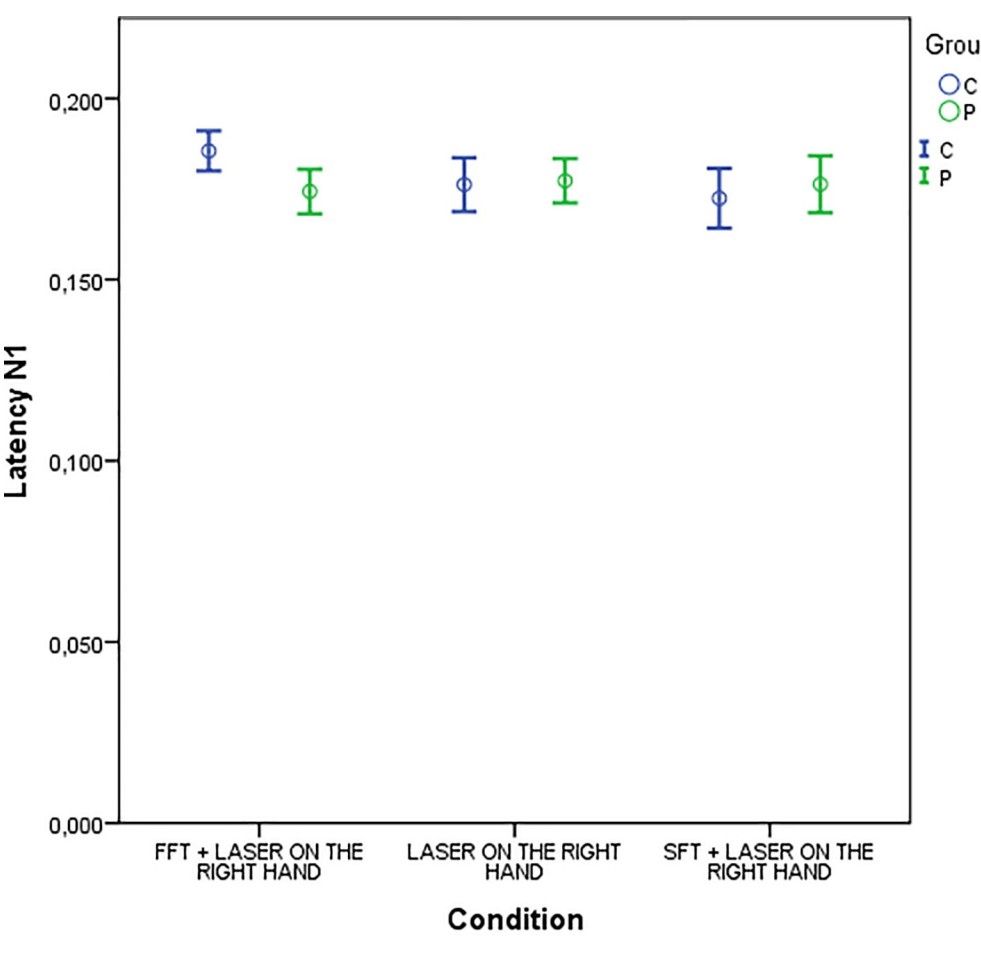

**Fig 9. Mean of N1 latency in experimental conditions with laser stimulation on the right hand for patients and controls.**

fast finger tapping, we can assume that the endogenous pain could contribute to the downregulation of the motor cortex activity in chronic patients. It would be an intrinsic feature of the disease, as it was independent from disease duration and severity.

**Table 10. Two-way ANOVA results.**

**Test of effects between subjects**

**Dependent variable: VAS**

| Source | Sum of squares III | Df | Mean squares | F | Sig. |
|---|---|---|---|---|---|
| Correct Model | 302079,314[b] | 17 | 17769,371 | 1,343 | ,164 |
| Intercept | 4221019,314 | 1 | 4221019,314 | 319,025 | ,000 |
| Group | 244266,783 | 2 | 122133,391 | 9,231 | ,000 |
| Condition | 2826,598 | 5 | 565,320 | ,043 | ,999 |
| Group * Condition | 15380,405 | 10 | 1538,040 | ,116 | 1,000 |
| Error | 4419156,555 | 334 | 13231,008 | | |
| Total | 73779578,000 | 352 | | | |
| Corrected total | 4721235,870 | 351 | | | |

VAS: dependent variable; Group and Conditions: factors.

## Effects of movement on laser evoked responses

Our results indicated that the amplitude of the LEPs components was different between FM patients and healthy subjects, independent from the different experimental conditions. According to previous studies [55, 30], patients with chronic pain can present alteration in expression of nociceptive responses. FM is characterised by a complex interaction of peripheral and central neuronal factors with a dysfunction of small fibers coexisting with central amplification of pain [56]. These phenomena could lead to variable group results, depending upon the prevailing phenotypical expression. In the present results, FM patients presented in basal with smaller LEP responses as compared to controls.

In general, movement seemed to affect LEPs in a not relevant way either in patients or in controls. Healthy subjects exhibited an increase of N1 latency during the execution of the fast finger tapping task. Probably this result is due to a possible movement–related somatosensory interference on cortical areas receiving multimodal somatosensory stimuli [57]. The features of N1 wave were unchanged during the other experimental conditions, suggesting that this interference could emerge only during fast movement. This phenomenon was absent in FM patients, as the reduced tone of motor cortex activation and the low motor performance could exert slight interference on concurrent cortical pain processing networks. If the lack of modulation effect of the finger tapping task on the amplitude of the laser cortical responses could be reasonable in FM for the low motor efficiency, the same phenomenon occurring in healthy subjects deserves further comments. Le Pera et al. [58] described the interaction between voluntary movement and LEPs changes in healthy volunteers. The N2P2 reduction occurred in the phase prior to the motor execution, when the laser stimulus was delivered on the hand that was supposed to move. In this case the process of movement preparation generated an inhibition of the vertex LEPs that was independent from a pure cognitive distraction effect. The process of movement preparation requires a cognitive commitment which could have an inhibitory function on pain. The vertex LEPs generate from the so-called salience matrix, which is a cortical network devoted to the arousal toward a relevant stimulus worthy of a behavioural motor response [59]. We can thus assume that the repetitive movement of finger tapping is not interpreted by the brain as a challenging task requiring pain silencing. Furthermore, the task of finger tapping in our case was not preceded by a warning stimulus able to put subjects in a condition of arousal and movement preparation which could in turn influence the areas involved in the complex processing of pain.

The effects of pain relief induced by experimental stimulation of the motor cortex could be due to the action of brain areas far from the site of stimulation [48]. Numerous studies [60] confirmed the analgesic effect induced by non-invasive stimulation of the motor cortex in patients with pain and specifically FM [61–62], though the level of evidence of their efficacy remains low [63]. In a previous study conducted in healthy controls and migraine patients, we observed that a single session of high frequency rTMS of the motor cortex reduced the LEP vertex complex in both groups, with a clear sham effect in migraine patients [64]. The motor cortex activation induced by finger tapping could not be able to reduce the pain-related cortical responses. The automatic and repetitive movement may thus exert a scarce modulation of cortical areas generating response to pain. This phenomenon is worthy of further confirmation as it could have a potential interest in the design of motor rehabilitation strategies.

Accordingly, the subjective pain intensity induced by laser stimulation, was unaffected by the concurrent execution of the finger tapping task.

## Conclusions

In conclusion, our results confirmed preliminary findings [21] about a dysfunction of motor cortex and impairment in motor speed in FM sufferers. A low tone of motor cortex activation could be intrinsic to FM and contribute to a scarce control on pain. In our experimental model, the concurrent phasic painful stimulation decreased motor cortex activation in healthy controls, confirming the inhibitory role of pain on motor cortex areas functions. The repetitive movement we used was unable to modulate the cortical responses to pain either in patients or in controls, a phenomenon which requires further confirmation, but of potential utility in rehabilitation strategy.

## Supporting information

**S1 Table. Latency and Amplitude of LEP components for patients' group.**
(DOCX)

**S2 Table. Latency and Amplitude of LEP components for controls' group.**
(DOCX)

**S3 Table. Correlations for FFT + LASER ON THE RIGHT HAND condition.**
(DOCX)

**S4 Table. Correlations for FFT + LASER ON THE LEFT HAND condition.**
(DOCX)

**S5 Table. Correlations for FFT condition.**
(DOCX)

**S6 Table. Correlations for SFT condition.**
(DOCX)

**S7 Table. Correlations for SFT + LASER ON THE LEFT HAND condition.**
(DOCX)

**S8 Table. Correlations for SFT + LASER ON THE RIGHT HAND condition.**
(DOCX)

**S9 Table. Correlations for baseline condition.**
(DOCX)

**S10 Table. Correlations for FFT condition.**
(DOCX)

**S11 Table. Correlations for FFT + LASER ON THE RIGHT HAND condition.**
(DOCX)

**S12 Table. Correlations for FFT + LASER ON THE LEFT HAND condition.**
(DOCX)

## Author Contributions

**Conceptualization:** Antonio Brunetti, Marina de Tommaso.

**Data curation:** Eleonora Gentile, Antonio Brunetti, Katia Ricci, Marina de Tommaso.

**Formal analysis:** Eleonora Gentile, Antonio Brunetti, Katia Ricci, Marina de Tommaso.

**Investigation:** Eleonora Gentile, Katia Ricci, Marianna Delussi, Vitoantonio Bevilacqua.

**Methodology:** Eleonora Gentile, Antonio Brunetti, Vitoantonio Bevilacqua, Marina de Tommaso.

**Project administration:** Marina de Tommaso.

**Supervision:** Vitoantonio Bevilacqua, Marina de Tommaso.

**Visualization:** Marina de Tommaso.

**Writing – original draft:** Eleonora Gentile, Antonio Brunetti, Marina de Tommaso.

**Writing – review & editing:** Eleonora Gentile, Antonio Brunetti, Vitoantonio Bevilacqua, Marina de Tommaso.

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
