## [Decision Letter · Decision Letter 0]

6 Nov 2019

PONE-D-19-19715

Mutual interaction between motor cortex activation and pain in Fibromyalgia: EEG-fNIRS study.

PLOS ONE

Dear Mrs Gentile,

Thank you for submitting your manuscript to PLOS ONE. After careful consideration, we feel that it has merit but does not fully meet PLOS ONE’s publication criteria as it currently stands. Therefore, we invite you to submit a revised version of the manuscript that addresses the points raised during the review process.

We would appreciate receiving your revised manuscript by Dec 21 2019 11:59PM. To enhance the reproducibility of your results, we recommend that if applicable you deposit your laboratory protocols in protocols.io, where a protocol can be assigned its own identifier (DOI) such that it can be cited independently in the future. For instructions see: http://journals.plos.org/plosone/s/submission-guidelines#loc-laboratory-protocols

We look forward to receiving your revised manuscript.

Kind regards,

Domenico Paolo Emanuele Margiotta

Academic Editor

PLOS ONE

Journal Requirements:

1. Thank you for including the following funding information within your acknowledgements section; "The study was supported by Bari Aldo Moro University Research funds."

The study was supported by Bari Aldo Moro University Research funds.

"The funders had no role in study design, data collection and analysis, decision to publish, or preparation of the manuscript"

Reviewers' comments:

Reviewer's Responses to Questions

**Comments to the Author**

1. Is the manuscript technically sound, and do the data support the conclusions?

Reviewer #1: Yes

Reviewer #2: Yes

2. Has the statistical analysis been performed appropriately and rigorously? 

Reviewer #1: Yes

Reviewer #2: Yes

3. Have the authors made all data underlying the findings in their manuscript fully available?

Reviewer #1: Yes

Reviewer #2: Yes

4. Is the manuscript presented in an intelligible fashion and written in standard English?

Reviewer #1: No

Reviewer #2: Yes

5. Review Comments to the Author

Reviewer #1: This is an interesting study exploring the relationship between motor activation and pain processing in fibromyalgia patients. This potentially represents a key point in fibromyalgia physiopathology. Methods are sound, the writing is straightforward.

I just have some suggestions:

- some paragraphs require a style revision since the meaning is not completely clear (471-475)

- there are other typos that need to be corrected

- The effect of motor activation on pain processing has a specific evolutionary purpose to allow fighting in spite of pain. Is it possible that finger tapping is not interpreted by the brain as a challenging task requiring pain silencing? please comment on this point.

- Another point deserving attention is the temporal evolution of motor dysfunction in fibromyalgia. Since no relation between clinical scales and fNIRS or LEPs were found in this patients, it could be speculated that the reduced motor activation is not the mere effect of chronic pain but it is a constitutional tract of the disease. To confirm this hypothesis the experimental paradigm should be applied in patients successfully treated with drugs such as duloxetine.

- In other conditions experiencing motor activation dysfunctions (i.e. stroke, aging), re-adaptive phenomena include a reduction of lateralization for the activation of ipsilateral motor pathways. Is it possible to evaluate this balancing in the present population? Does the pain cortical activation follow the same reorganization?

Reviewer #2: GENERAL COMMENTS 

Dr Gentile and Colleagues provide this original article on the complex mechanisms of interaction between motor and pain. The study was conducted performing slow and fast finger tapping task alone and in concomitant with nociceptive laser stimulation. The major strength of this work is that the authors used simultaneous multimodal method of laser-evoked potentials and functional near-infrared spectroscopy. The results show a dysfunction of motor cortex and impairment in motor speed in patients affected with fibromyalgia. Those data must be studied in depth for the important implications that they could have in the rehabilitation field.

SPECIFIC COMMENTS

- Line 31: FNIRS was not defined before

- In all the text Fibromyalgia should be abbreviated as FM

- Line 62: the dot after study should be eliminated

- Materials and method – Subject: which criteria were used for FM classification?

- In Experimental study design there are some concept repetitions (line 112 and line 121)

- Line 407: what does FT mean?

6. PLOS authors have the option to publish the peer review history of their article (what does this mean?). If published, this will include your full peer review and any attached files.

Reviewer #1: No

Reviewer #2: No

---

## [Author Response · Author response to Decision Letter 0]

12 Dec 2019

Dear reviewers, 

We are pleased for your comments of this article. I attach the corrections about your suggestion.

Reviewer #1

This is an interesting study exploring the relationship between motor activation and pain processing in fibromyalgia patients. This potentially represents a key point in fibromyalgia physiopathology. Methods are sound, the writing is straightforward.

I just have some suggestions:

- Some paragraphs require a style revision since the meaning is not completely clear (471-475)

- There are other typos that need to be corrected.

- The effect of motor activation on pain processing has a specific evolutionary purpose to allow fighting in spite of pain. Is it possible that finger tapping is not interpreted by the brain as a challenging task requiring pain silencing? please comment on this point.

We thank the reviewer for his useful comments. We completely agree with your suggestion and included this concept in the revised discussion. The changes were outlined in green color.

-Another point deserving attention is the temporal evolution of motor dysfunction in fibromyalgia. Since no relation between clinical scales and fNIRS or LEPs were found in these patients, it could be speculated that the reduced motor activation is not the mere effect of chronic pain but it is a constitutional tract of the disease. To confirm this hypothesis the experimental paradigm should be applied in patients successfully treated with drugs such as duloxetine.

We have not yet concluded to evaluate patients at follow up, but it will be a matter of a future study. We integrated this in the discussion.

- In other conditions experiencing motor activation dysfunctions (i.e. stroke, aging), re-adaptive phenomena include a reduction of lateralization for the activation of ipsilateral motor pathways. Is it possible to evaluate this balancing in the present population? Does the pain cortical activation follow the same reorganization?

We computed the lateralization index and we obtained a similar mechanism for the activation on the motor areas between groups. We integrated the results and comments in the paper.

Reviewer #2: 

-Line 31: FNIRS was not defined before

- In all the text Fibromyalgia should be abbreviated as FM

- Line 62: the dot after study should be eliminated

- In Experimental study design there are some concept repetitions (line 112 and line 121)

- Line 407: what does FT mean?

We thank the reviewer for his valuable suggestions. The changes were outlined in yellow color.

-Materials and method – Subject: which criteria were used for FM classification?

Thank you for the consideration of our work. We added a more detailed description of ACR 2010 criteria.

---

## [Editor Report · Decision Letter 1]

9 Jan 2020

Mutual interaction between motor cortex activation and pain in Fibromyalgia: EEG-fNIRS study.

PONE-D-19-19715R1

Dear Dr. Gentile,

We are pleased to inform you that your manuscript has been judged scientifically suitable for publication and will be formally accepted for publication once it complies with all outstanding technical requirements.

With kind regards,

Domenico Paolo Emanuele Margiotta

Academic Editor

PLOS ONE

---

## [Editor Report · Acceptance letter]

14 Jan 2020

PONE-D-19-19715R1 

Mutual interaction between motor cortex activation and pain in Fibromyalgia: EEG-fNIRS study. 

Dear Dr. Gentile:

I am pleased to inform you that your manuscript has been deemed suitable for publication in PLOS ONE. Congratulations! Your manuscript is now with our production department. 

With kind regards,

on behalf of

Dr. Domenico Paolo Emanuele Margiotta 

Academic Editor

PLOS ONE